# Modeling of Ship DC Power Grid and Research on Secondary Control Strategy

**Hong Zeng *** 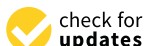**, Yuanhao Zhao, Xuming Wang, Taishan He and Jundong Zhang**

Marine Engineering College, Dalian Maritime University, Dalian 116026, China
* Correspondence: zenghong@dlmu.edu.cn

**Abstract:** Compared to alternating current (AC) grids, direct current (DC) grids are becoming more and more popular. A power distribution approach is suggested in order to solve the issue of uneven power distribution of distributed generation (DG) in a ship DC microgrid. Power control is carried out using a tracking differentiator (TD), while the output power change rate is not greater than the maximum power ramp rate permitted by the battery, and state-of-charge balance is attained quickly. The proposed strategy also reduces the communication pressure on the power grid. A distributed hierarchical control model of a DC microgrid based on a consensus algorithm is created in order to validate the suggested methodology. The simulation results demonstrate that the established model is capable of simulating the DC microgrid accurately, that the states of charge values of the five batteries gradually converge under the adjustment of the secondary strategy, and that the suggested strategy is reasonable and efficient.

**Keywords:** ship DC microgrid; power distribution; tracking differentiator; consensus algorithm

## 1. Introduction

A microgrid is a self-sufficient, sustainable system that uses distributed generation and blends distributed renewable energy with local electrical loads [1]. Due to its benefits of flexible operation, no synchronization issue with generator on/off, high power density, and space saving, marine DC microgrid has grown in popularity recently [2–5]. Fuel consumption can be decreased by integrating the energy storage device into the DC grid easily [5]. The method of paralleling converters is frequently employed, since powering the DC grid with a single converter will make the system hotter and shorten its lifespan. There are also several studies on parallel DC–DC converters [6–10].

The energy storage module typically uses voltage control, and the output properties resemble those of the voltage source. Running in parallel, each converter's different output line impedance results in a significant variance in output current, which has an impact on the power equalization between the energy storage modules. In order to solve this issue, virtual resistors are added to distributed energy sources (DERs) in [11] in order to increase the accuracy of output power sharing. However, the proposed solution is only appropriate for the dedicated lines of each DER to be connected directly to the point of common coupling (PCC), and the load is only connected to the PCC, which is inconsistent with the actual distribution system. In [12–14], hierarchical control is recommended, where the higher layer employs secondary control in order to compensate for the bus voltage drop and maintain the bus voltage stability, while the bottom layer uses droop control to accomplish precise current distribution. The droop curve was converted to a parabola by Liu Haiyuan [15], who also suggested a two-factor adaptive droop approach. On the basis of the conventional droop control, the droop curve was transformed from a straight line into a curve with a variable slope by raising the power factor. Under the premise that communication is not added, the contradiction between the power distribution characteristics and the DC bus voltage quality is resolved, and the DC bus voltage quality is enhanced. However, the DC bus voltage will also deviate significantly when the output current exceeds

the rated value. A parallel current sharing control strategy based on frequency injection was proposed in [16], which adjusts the output voltage of each distributed power source in accordance with the reactive power generated by the AC signal. This strategy successfully solves the issue of power distribution brought on by differences in line impedance. The injection of the AC signal will, however, increase the output voltage ripple, which will have an impact on the power quality. In order to address the issue of voltage ripple, Zhang Qinjin [17] suggested a DC microgrid power distribution method based on active frequency injection, adding a mode switching segment to the parallel current sharing control system based on frequency injection, and the mode is ordinary droop control in steady state to solve the problem of voltage ripple. Mehdi Baharizadeh [18] proposed a two-layer control scheme based on P-dv/dt droop characteristics for precise power sharing and voltage regulation of DC microgrids. The trade-off between damping and sharing accuracy must be taken into account while setting the rappel factor. Wen Wang [19] improved the droop control based on the battery state of charge; the droop coefficient is inversely proportional to the n-time power of the SOC, which realizes the balance of the SOC during the battery charging and discharging process under the condition of no communication. In [20], each DER communicates with the central controller through a separate bidirectional communication link in the hierarchical control. This link is utilized for both the transmission of DER-specific control commands from the central controller to the DER and the transmission of measured signals from the DER to the central controller. Therefore, the total communication bandwidth of the central controller (equal to the sum of the bandwidths of DERs) is much higher than that of traditional secondary control schemes [21]. Centralized secondary control can eliminate voltage deviation and current sharing error. However, there is a single point of failure (SPOF) problem in the central controller [13]. Additionally, since a communication link from the central control to each local unit is necessary, communication costs are higher, and the reliability and flexibility are reduced as a result. This work takes a fully distributed control strategy, which means that there are no single points of failure and only communication links are needed between nearby units. Plug and play functionality enhances system flexibility.

In previous distributed control strategies, in order to compensate for the voltage drop, devices need to exchange their own voltage information with adjacent devices. To achieve accurate current sharing, the equipment needed to exchange its own current information with adjacent equipment. In order to balance the state of charge of the battery, the device needs to exchange one or two messages with adjacent devices. In the control strategy proposed in this paper, the device only needs to exchange one or two messages with adjacent devices. After sharing data, the unit modifies its output current through the tracking differentiator while the output power is constrained by its power ramp rate, so that its own state of charge (SOC) closely tracks the average state of charge ($SOC_{avg}$) of all units. The four-phase interleaved parallel circuit has the advantages of a simple structure principle and small volume, and the mathematical model after coordinate transformation is similar to the single-phase model. This circuit is used in the energy storage booster circuit.

The rest of this paper is organized as follows: Section 2 introduces the converter and its controller; Section 3 introduces the proposed strategy; Section 4 introduces the consensus algorithm and analyzes the dynamics and convergence of the algorithm; Section 5 simulates and analyses the feasibility of the proposed strategy; Section 6 summarizes the paper.

## 2. Converter and Its Controller

### 2.1. Converter

The boost/buck circuit adopts four-phase interleaved parallel bidirectional converter. Formula (1) is the state average model. As shown in Figure 1. $d_1$, $d_2$, $d_3$ and $d_4$ are the duty ratios of $Q_2$, $Q_4$, $Q_6$ and $Q_8$, respectively. $d'_i = 1 - d_i$. $v_1$ is the voltage at the low voltage side of the converter, $v_o$ is the voltage at the high voltage side of the converter, $L$ is the

inductance, $M$ is the mutual inductance, $r$ is the resistance, and $C$ is the output capacitance of the converter.

$$
\begin{cases}
v_1 - d_1' v_o = 2L\frac{di_{L1}}{dt} - r \cdot i_{L1} + M\frac{di_{L2}}{dt} + M\frac{di_{L4}}{dt} \\
v_1 - d_2' v_o = M\frac{di_{L1}}{dt} - r \cdot i_{L2} + 2L\frac{i_{L2}}{dt} + M\frac{di_{L3}}{dt} \\
v_1 - d_3' v_o = M\frac{di_{L2}}{dt} - r \cdot i_{L3} + 2L\frac{di_{L3}}{dt} + M\frac{di_{L4}}{dt} \\
v_1 - d_4' v_o = M\frac{di_{L1}}{dt} - r \cdot i_{L4} + M\frac{di_{L3}}{dt} + 2L\frac{i_{L4}}{dt} \\
C\frac{dv_o}{dt} = d_1' i_{L1} + d_2' i_{L2} + d_3' i_{L3} + d_4' i_{L4} - \frac{v_o}{R_L}
\end{cases} . \tag{1}
$$

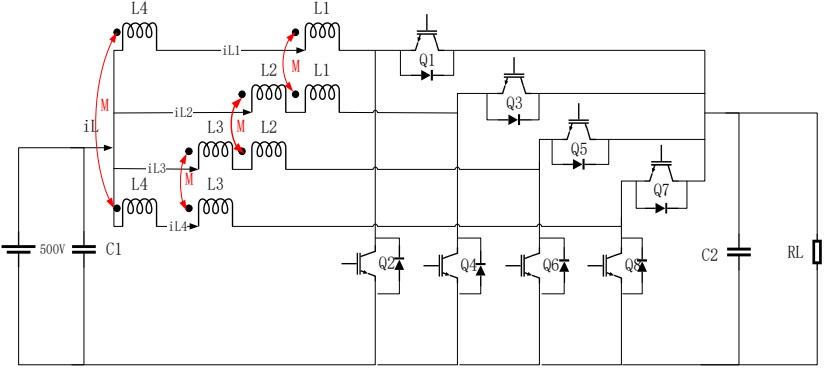

**Figure 1.** Four-phase interleaved parallel bidirectional converter.

### 2.2. Controller

The switching frequency $fs$ of the IGBT is 5 kHz. The cut-off frequency of the current open loop is set to 1000 Hz. The open-loop cut-off frequency of the voltage loop is set to 100 Hz. The carrier of the drive circuit is staggered by 1/4 period in sequence. The controller design is shown in Figure 2. $v_o^*$ is the given value of the converter output voltage. The detailed derivation process is shown in Appendix A.

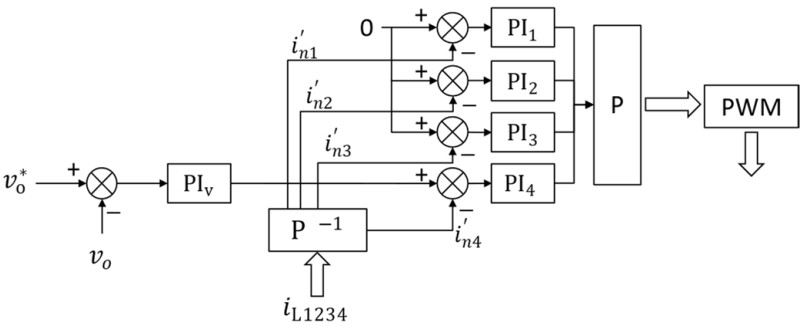

**Figure 2.** Controller Diagram.

## 3. Power Distribution Control Strategy

### 3.1. Traditional Droop Control

The main forms of traditional DC droop control are I-V droop control and P-V droop control [22], which are expressed as:

$$
v_o^* = V_{ref} - R_v i_o \tag{2}
$$

$$
u_o^* = U_{ref} - k_p P_o \tag{3}
$$

where $V_{ref}$ is the reference value of the DC bus voltage; $v_o^*$ is the reference value of the output voltage of the converter; $R_v$ is the droop coefficient of the converter; $i_o$ is the output current of the converter; $U_{ref}$ is the reference value of the DC bus voltage; $u_o^*$ is the reference value of the output voltage of the converter; $P_o$ is the output power of the converter; and $k_p$ is the droop coefficient in the P-V droop control.

It can be seen from Formulas (2) and (3) that a voltage drop occurs in droop control, which is an inherent defect. To ensure system stability, the drop coefficient is required to satisfy the following constraints:

$$0 < R_v < \frac{\Delta V_{o\max}}{i_{o\max}} \tag{4}$$

$$0 < k_P \leq \frac{\Delta U_{o\max}}{P_{\max}} \tag{5}$$

where $\Delta V_{o\max}$ is the maximum allowable deviation of the bus voltage, and $i_{o\max}$ is the maximum allowable output current of the converter. $\Delta U_{o\max}$ is the maximum allowable deviation of the bus voltage, and $P_{\max}$ is the maximum allowable output power of the converter.

Also, the output current or output power cannot be scaled accurately due to inconsistent line impedances. As shown in Figure 3, $R_{dg}$, R1, R2 ... Rn are the line impedance of generator and batteries, respectively.

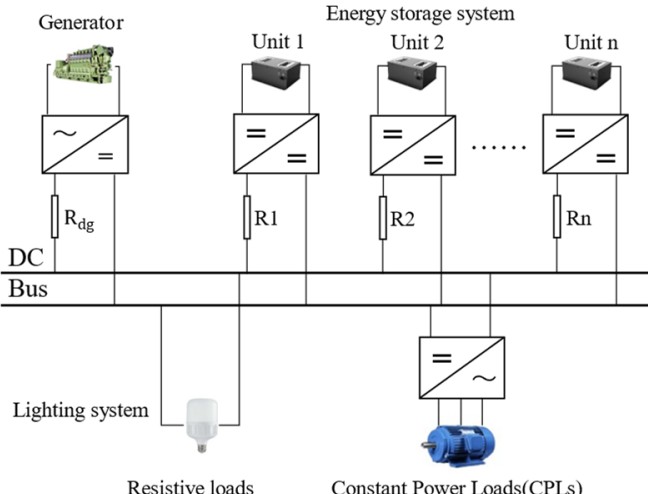

**Figure 3.** Physical layer structure diagram of ship DC microgrid.

Selecting a smaller droop curve coefficient can reduce the voltage deviation but will reduce the accuracy of shared current. Selecting a larger droop curve coefficient can improve the accuracy of the shared current but will increase the voltage deviation. Therefore, the traditional current droop control has an inherent contradiction in the pursuit of small voltage deviation and large current sharing. This is the disadvantage of current droop control.

*3.2. Distributed Hierarchical Control Scheme Based on Consensus Algorithm*

To avoid voltage deviation, load current cannot be accurately shared in addition to other problems. Secondary control is adopted.

$$\delta v_I = \left( \frac{k_{isc}}{s} + k_{psc} \right) \cdot \left( \overline{i_o} - i_{oi} \right) \tag{6}$$

$$\delta v_o = \left( \frac{k_{isv}}{s} + k_{psv} \right) \cdot \left( v^* - \overline{v}_o \right) \tag{7}$$

where $\delta v_I$ and $\delta v_o$ are the compensation term of the shared current and the voltage recovery, respectively. $i_o$ and $v_o$ are the average output current and the average output voltage of all energy storage units, which are obtained by consensus algorithm. $k_{isc}$ and $k_{psc}$ are integral and proportional terms of the shared current controller, and $k_{isv}$ and $k_{psv}$ are inte-

gral and proportional terms of the voltage recovery controller. The sum of $\delta v_I$ and $\delta v_o$ is sent to the main controller.

At the top level of the controller, a consensus algorithm is used to realize information sharing and averaging among distributed agents. This helps to find the average value $i_o$ of the total generation current and the average value $v_o$ of the output voltage [13].

A new secondary control strategy can combine the voltage drop compensation regulation and the shared current regulation into one. Adjacent devices only need to exchange one unit of information, and only one PI regulator is needed.

$$\xi_i = \gamma_i v_{oi} \tag{8}$$

$$\gamma_i = 1 - k\frac{i_{oi}}{i_{o\max}} \tag{9}$$

where $v_{oi}$ is the output voltage of the $i$-th converter. $i_{oi}$ is the output current of the $i$-th converter. $i_{o\max}$ is the maximum output current of all converters. $k$ is the gain ($0 < k < 1$).

The PI regulator output $\delta v$ is

$$\delta v = \left(\frac{k_i}{s} + k_{\mathrm{p}}\right) \cdot \left(V_{ref} - \frac{\xi_{avg}}{\gamma_i}\right) \tag{10}$$

$$\xi_{avg} = \frac{\sum_{i=1}^{n}\xi_i}{n} \tag{11}$$

where $k_i$ and $k_p$ are integral and proportional terms of the PI controller.

$$V_{ref} - \frac{\xi_{avg}}{\gamma_i} = \frac{V_{ref}}{n}\left(1 - \frac{v_{oi}}{V_{ref}} + \sum_{j=1,j\neq i}^{n}1 - \frac{v_{oj}}{V_{ref}}\frac{\gamma_j}{\gamma_i}\right) \tag{12}$$

Under the regulation of PI controller:

$$\gamma_1 = \gamma_2 = \ldots = \gamma_n \tag{13}$$

$$V_{ref} = \overline{v} \tag{14}$$

As shown in Figure 4, for unit $i$, the average value $SOC_{avg}$ is obtained by communicating with adjacent devices through consensus algorithm. Through the tracking differentiator, $SOC_i$ tracks $SOC_{avg}$, the output speed is the change of battery output power $\Delta P$, and the maximum power ramp rate allowed by the battery is the adjustment coefficient of the tracking differentiator [23].

$$\gamma_i = 1 - k\frac{i_{oi} + \Delta i_i}{i_{o\max}} \tag{15}$$

where $\Delta i_i = \Delta P / v_{oi}$.

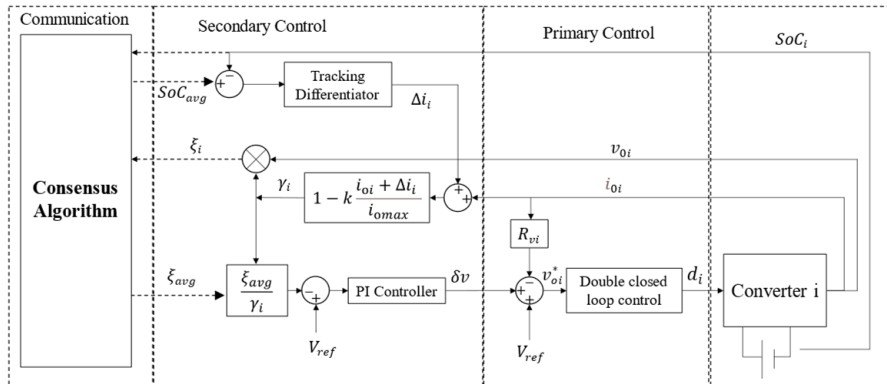

**Figure 4.** Hierarchical control diagram.

### 3.3. Stability Analysis

Based on [24], the model shown in Figure 5 is established for stability analysis. In the model, I regulator is used as the secondary controller.

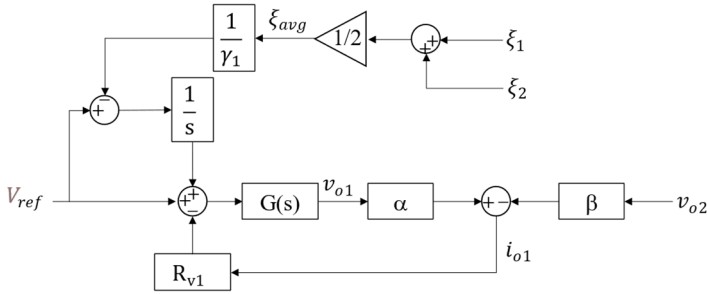

**Figure 5.** Equivalent model for stability analysis.

In Figure 5, $G(s)$ is the voltage loop transfer function, which can be approximately equal to 1. Reference [24] for converter output voltage stability analysis and derivation process. This paper analyzes the current stability. According to Figure 5, the following formula is derived:

$$v_{o1} = \left[ V_{ref} - R_{v1}i_{o1} + \frac{1}{s}(V_{ref} - \frac{\xi_{avg}}{\gamma_1}) \right] G(s) \tag{16}$$

In addition,

$$v_{o1} = (i_{o1} + i_{o2})R_L + i_{o1}R_{Line1} \tag{17}$$

$$v_{o2} = (i_{o1} + i_{o2})R_L + i_{o2}R_{Line2} \tag{18}$$

where $R_{Line1}$ and $R_{Line2}$ are output impedances of converter 1 and 2, respectively.

Substitute Formulas (17) and (18) into Formula (16).

$$i_{o1} = \frac{\Delta i_{o1}\frac{k}{i_{o\max}}\left\{ i_{o1}\left[ (\frac{s}{G(s)}+1/2)(R_L+R_{Line1})+R_{v1}s \right]-V_{ref}(s+1) \right\}+V_{ref}(s+1)-i_{o2}[(\frac{s}{G(s)}+1/2)R_L\gamma_1+(R_L+R_{Line2})\gamma_2/2]}{s\left\{ [\frac{1}{G(s)}(R_L+R_{Line1})+R_{v1}](1-\frac{k}{i_{o\max}}i_{o1})+V_{ref}\frac{k}{i_{o\max}} \right\}+\frac{1}{2}(R_L+R_{Line1})(1-\frac{k}{i_{o\max}}i_{o1})+\frac{R_L\gamma_2}{2}+V_{ref}\frac{k}{i_{o\max}}} \tag{19}$$

Linearize the variables of Formula (19) with small signal model.

$$\left. \frac{\hat{i}_{o1}}{\Delta\hat{i}_{o1}} \right|_{\hat{i}_{o2}=0} = \frac{\frac{k}{i_{o\max}}\left\{ i_{o1}\left[ (\frac{s}{G(s)}+1/2)(R_L+R_{Line1})+R_{v1}s \right]-V_{ref}(s+1) \right\}}{s\left\{ \left[ \frac{1}{G(s)}(R_L+R_{Line1})+R_{v1} \right](1-\frac{k}{i_{o\max}}i_{o1})+V_{ref}\frac{k}{i_{o\max}} \right\}+\frac{1}{2}(R_L+R_{Line1})(1-\frac{k}{i_{o\max}}i_{o1})+\frac{R_L\gamma_2}{2}+V_{ref}\frac{k}{i_{o\max}}} \tag{20}$$

The pole of the transfer function is in the stable region of the s plane, and the output current is stable.

## 4. Consensus Algorithm

### 4.1. Dynamic Consensus Algorithm

The basic consensus algorithm using continuous-time (CT) and discrete-time (DT) integrator agents can be described as [25,26].

$$\dot{x}_i(t) = \sum_{j\in N_i} a_{ij} \cdot (x_j(t) - x_i(t)) \tag{21}$$

$$x_i(k+1) = x_i(k) + \varepsilon \cdot \sum_{j\in N_i} a_{ij} \cdot (x_j(k) - x_i(k)) \tag{22}$$

where $i = 1, 2, \ldots, N_T$, $N_T$ is the total number of agent nodes. $x_i$ is the state of agent $i$. $a_{ij}$ indicates the connection status between node $i$ and node $j$, $a_{ij} = 0$ if there is no link between

them. $N_i$ is the set of indexes of the agents that can be connected with agent $i$, and $\varepsilon$ is the constant edge weight that is used to adjust DCA's dynamic characteristics.

In this paper, the discrete time (DT) form of the consensus algorithm (22) is employed due to the discreteness of communication data transfer. Furthermore, an improved algorithm known as DCA [27] is used to guarantee the accurate consensus in dynamically changing situations.

$$x_i(k+1) \;=\; x_i(0) + \varepsilon \cdot \sum_{j \in N_i} \delta_{ij}(k+1) \tag{23}$$

$$\delta_{ij}(k+1) \;=\; \delta_{ij}(k) + a_{ij} \cdot \big(x_j(k) - x_i(k)\big) \tag{24}$$

where $\delta_{ij}(k)$ stores the cumulative difference between the two agents. $\delta_{ij}(0) \;=\; 0$. Based on Formulas (23) and (24), it is obvious that the ultimate consensus value depends on the initial value $x_i(0)$ and that the algorithm will converge to the right average regardless of changes in that value $x_i(0)$.

From the perspective of the system, the iterative algorithm's vector form can be stated as [25,26].

$$x(k+1) \;=\; W \cdot x(k) \tag{25}$$

where $x$ is the state vector $x(k) = [x1(k),\ x2(k),\ \dots,\ xN_T(k)]^T$, and $W$ is the weight matrix of the communication network.

If a constant edge weight $\varepsilon$ is considered, $W$ can be described as

$$W \;=\; I - \varepsilon \cdot L \tag{26}$$

$$L \;=\; \begin{bmatrix} \displaystyle\sum_{j \in N_1} a_{1j} & \cdots & -a_{1N_T} \\ \vdots & \ddots & \vdots \\ -a_{1N_T} & \cdots & \displaystyle\sum_{j \in N_T} a_{N_T j} \end{bmatrix} \tag{27}$$

where $L$ is the Laplacian matrix of the communication network [28,29], $N_i$ is the set of indexes of the agents that are connected with agent $I$, and $N_T$ is the total number of agents.

The final consensus equilibrium $x_{eq}$ is

$$x_{eq} \;=\; \lim_{k \to \infty} x(k) \;=\; \lim_{k \to \infty} W^k x(0) \;=\; \left(\frac{1}{N_T} \mathbf{1} \cdot \mathbf{1}^T\right) x(0) \tag{28}$$

where $x(0) = [x1(0), x2(0), \dots, x_{N_T}(0)]$ is the vector of the initial values held by each agent, 1 denotes a vector where all the components equal one. The detailed proof of the algorithm convergence can be found in [25]. In this paper, the initial value is the locally measured state of charge, output current, and voltage.

### 4.2. Algorithm Convergence and Dynamic

In order to guarantee the stability and fast convergence of the communication algorithm, $\varepsilon$ must be properly chosen. The fastest rate problem is known as the "the symmetric fastest distributed linear averaging" (symmetric FDLA) problem, presuming that the communication link is bidirectional. With specific restrictions on the weight matrix $W$, this issue is essentially the minimum of the spectral radius of the matrix $W - (1/N_T)\cdot\mathbf{1} \cdot \mathbf{1}^T$. When the requirements listed below are satisfied [30], the fastest convergence rate is attained:

$$\varepsilon \;=\; \frac{2}{\lambda_1(L) + \lambda_{n-1}(L)} \tag{29}$$

where $\lambda_j(\cdot)$ is the symmetric matrix's $j$-th biggest eigenvalue. The eigenvalues of $L$ are $[0, 1.382, 1.382, 3.618, 3.618]^T$ according to the ring's topology, which results in the best $\varepsilon = 2/5$. Figure 6 shows a comparison of the convergence rates as an illustration. The consensus algorithm's sample period in this instance is set to $T_{ca} = 100$ ms. Starting at $x(0) = [1,$

2, 3, 4, 5], the system eventually converges to a mean of 3. In the results shown in Figure 6, the constant edge weight $\varepsilon$ has a significant impact on the dynamics of DCA. When $\varepsilon = 2/5$, the spectral radius $\rho(W - (1/N_T) \cdot \mathbf{1} \cdot \mathbf{1}^T)$ is the lowest and the fastest transient response is achieved.

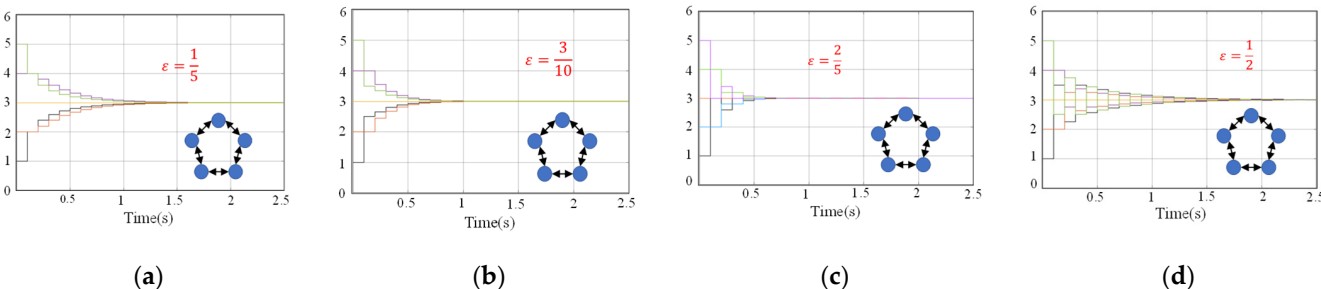

(**a**)       (**b**)       (**c**)       (**d**)

**Figure 6.** Comparison of convergence rates under different weights. (**a**) $\varepsilon = 0.2$; (**b**) $\varepsilon = 0.3$; (**c**) $\varepsilon = 0.4$; (**d**) $\varepsilon = 0.5$.

The communication network's structure has an effect on the system dynamics, as seen in Figure 7. Four topologies were taken into account in this analysis: (1) line-shaped; (2) ring-shaped; (3) star-shaped; and (4) fully connected networks. When the value of $\varepsilon$ is set in accordance with Formula (21), the fastest rate of convergence is attained in all cases (the optimum value for each case is shown in the figure). The results demonstrated that the network dynamics are significantly influenced by the communication topology. Undoubtedly, a full connected network offers the quickest convergence, but in most situations, it is not viable due to the high cost of network communication.

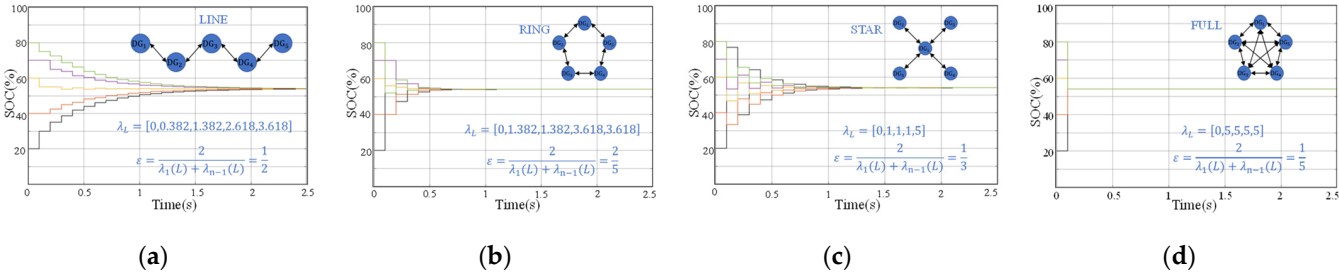

(**a**)       (**b**)       (**c**)       (**d**)

**Figure 7.** Network dynamics under different topologies. (**a**) Line; (**b**) ring; (**c**) star; (**d**) full.

## 5. Simulation and Analysis

Refer to 《IEEE Recommended Practice for 1 kV to 35 kV Medium-Voltage DC Power Systems on Ships》 and the actual case information for DC ships in the literature [31]. The model was established in MATLAB/Simulink, and the proposed method was simulated and verified with a simulation step size of $5 \times 10^{-5}$ s. Additionally, the line impedance values are as follows: R1 = 0.08 $\Omega$, R2 = 0.07 $\Omega$, R3 = 0.06 $\Omega$, R4 = 0.04 $\Omega$, and R5 = 0.02 $\Omega$. Droop coefficient Rv is set to 0.5. The rest of the simulation circuit parameters are shown in Table 1.

Figure 8b shows the current distribution of the converters when the secondary strategy is not used. Before the 4th second, the minimum and maximum output current of the converters differ by about 5A. At the 4th second, when the load is increased, the difference between the minimum and maximum output current of the converter is about 8A. It can be seen that with the increase of the load, the output current of the converters has a larger and larger difference. The unequal output current of the converters will inevitably lead to a different state of charge of the batteries. Figure 8a,c shows that when the output current of the converter increases, the output voltage of the converter will decrease, which results in more deviation of the DC bus voltage from the rated value.

**Table 1.** Circuit and control parameters.

| Device Parameters/Units | Value | Control Parameters/Units | Value |
|---|---|---|---|
| Inductance L/mH | 2.5 | TD Adjustment coefficient h0 | $1 \times 10^{-3}$ |
| Mutual inductance M/mH | $-2$ | TD filter coefficient r0 | $1 \times 10^{3}$ |
| Output capacitance C2/μF | 1000 | Secondary control kp | 0 |
| Battery voltage v1/V | 500 | Secondary control ki | 1 |
| Battery capacity Sn/kwh | 2 | Communication latency Tca/ms | 100 |
| Maximum output current of converter $i_{omax}$/A | 100 | Bus Voltage/V | $1000 \pm 10\%$ |

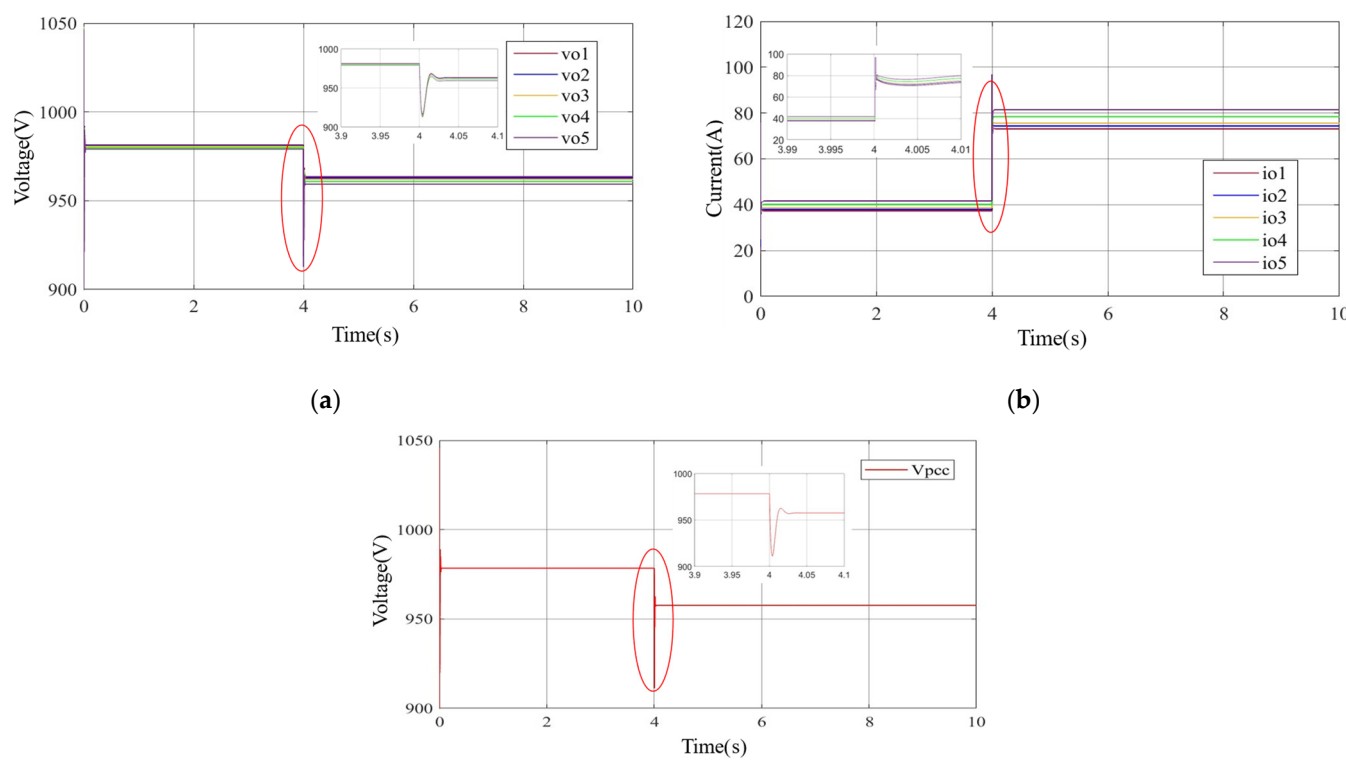

**Figure 8.** Simulation without secondary control: (**a**) output voltages of converters; (**b**) output currents of converters; (**c**) DC bus voltage.

Simulation analysis under the proposed strategy.

In this simulation, a generator is connected to the power grid through a rectifier converter and works together with the battery pack. The dq model is adopted as a three-phase voltage source. As shown in Figure 9a, the initial states of charge of the five batteries are different; they are 70%, 65%, 60%, 55% and 50%, respectively. At t = 5 s, the secondary control starts. As shown in Figure 9b, the output currents of converters gradually converge. As shown in Figure 9c, the output voltages of converters increase. As shown in Figure 9d, the DC bus voltage gradually approaches the rated value. At t = 15 s, the battery exchanges its own state of charge information with adjacent devices. As shown in Figure 9a, under the adjustment of the proposed strategy, the states of charge of the batteries gradually converge. At about 35 s, the output currents of converters do not change, because some constraints are added in the tracking differentiator program, such as limiting its output value. At t = 40 s, the generator increases the output power until the 60th second, as shown in Figure 9b, and the battery pack gradually changes from discharging to charging. At t = 70 s, as shown in Figure 9a, the size of the state of charge of the five batteries is almost

the same, and the difference is about 0%. As shown in Figure 9b, at t = 75 s, the output currents of the battery pack converge again. As shown in Figure 9d, the bus voltage drop is less than 5V. When the generator output power is stable, the bus voltage remains stable under the regulation of the proposed strategy.

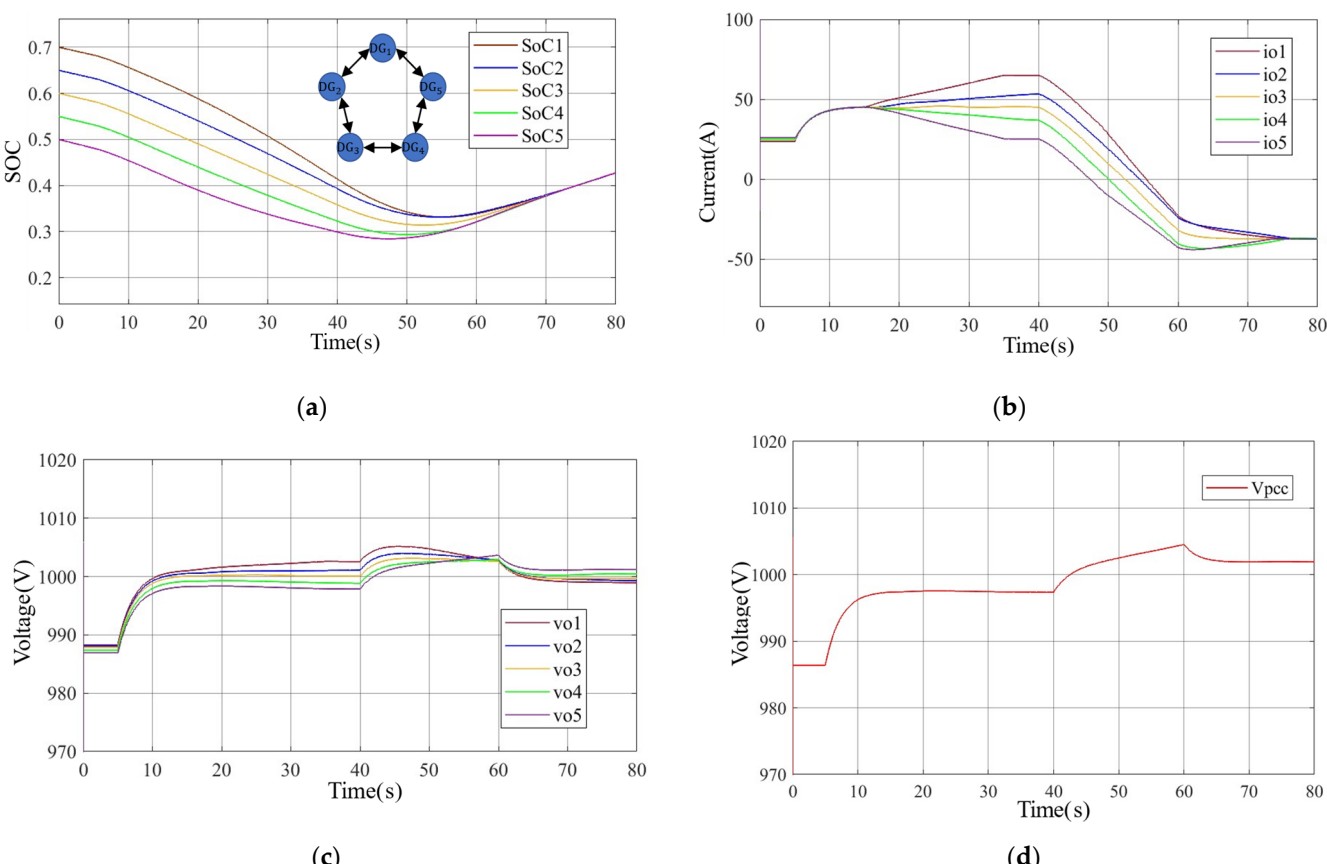

**Figure 9.** Simulation under the proposed strategy: (**a**) state of charge of batteries; (**b**) output current of converters under secondary control; (**c**) output voltage of converters under secondary control; (**d**) DC bus voltage under secondary control.

## 6. Conclusions

In this paper, a ship DC microgrid model is established which consists of three parts: equipment, controller, and communication network. Under the conditions of having/not having a secondary control strategy, we simulated the operation of the ship DC microgrid, and made a comparative analysis. The simulation results show that, under the adjustment of the proposed strategy, the problem of an unbalanced state of charge of the energy storage module is solved. After solving this problem, the state of charge adjustment strategy stops running, i.e., $\Delta i_i = 0$. In addition, under the regulation of the proposed secondary control strategy, adjacent devices only need to exchange two messages. After adjusting the state of charge of the battery pack, adjacent devices only need to exchange one message, reducing the communication pressure of the ship DC microgrid.

**Author Contributions:** Conceptualization, Y.Z. and H.Z.; methodology, Y.Z.; software, Y.Z.; formal analysis, Y.Z.; investigation, Y.Z. and T.H.; resources, Y.Z. and J.Z.; writing—original draft preparation, Y.Z. and H.Z.; and writing—review and editing, Y.Z., X.W., H.Z., Y.Z., X.W., T.H. and J.Z. All authors have read and agreed to the published version of the manuscript.

**Funding:** This research was supported by a grant from the High Technology Ship Research and Development Program of the Ministry of Industry and Information Technology of China (CJ02N20).

**Institutional Review Board Statement:** Not applicable.

**Informed Consent Statement:** Not applicable.

**Data Availability Statement:** Not applicable.

**Conflicts of Interest:** The authors declare no conflict of interest.

## Appendix A

Converter model derivation.

(1)  Mathematical decoupling

Write the Formula system (1) in matrix form as follows:

$$
v_1 \begin{bmatrix} 1 \\ 1 \\ 1 \\ 1 \end{bmatrix} - v_o \begin{bmatrix} d'_1 \\ d'_2 \\ d'_3 \\ d'_4 \end{bmatrix} = \begin{bmatrix} r+2Ls & Ms & 0 & Ms \\ Ms & r+2Ls & Ms & 0 \\ 0 & Ms & r+2Ls & Ms \\ Ms & 0 & Ms & r+2Ls \end{bmatrix} \begin{bmatrix} i_{L1} \\ i_{L2} \\ i_{L3} \\ i_{L4} \end{bmatrix} \tag{A1}
$$

$V_L$ is the voltage across the inductor, and $I_L$ is the current flowing through the inductor. Their relationship is as shown in Formula (A2):

$$
V_L(s) = Z(s)I_L(s) \tag{A2}
$$

$Z(s)$ is the equivalent impedance.

$$
Z(s) = \begin{bmatrix} r+2Ls & Ms & 0 & Ms \\ Ms & r+2Ls & Ms & 0 \\ 0 & Ms & r+2Ls & Ms \\ Ms & 0 & Ms & r+2Ls \end{bmatrix} \tag{A3}
$$

Decouple $Z(s)$.

$$
H(s) = P^{-1}Z(s)P \tag{A4}
$$

$$
P^{-1} = \begin{bmatrix} -\frac{1}{2} & 0 & \frac{1}{2} & 0 \\ 0 & -\frac{1}{2} & 0 & \frac{1}{2} \\ -\frac{1}{4} & \frac{1}{4} & -\frac{1}{4} & \frac{1}{4} \\ \frac{1}{4} & \frac{1}{4} & \frac{1}{4} & \frac{1}{4} \end{bmatrix} \tag{A5}
$$

$$
P = \begin{bmatrix} -1 & 0 & -1 & 1 \\ 0 & -1 & 1 & 1 \\ 1 & 0 & -1 & 1 \\ 0 & 1 & 1 & 1 \end{bmatrix} \tag{A6}
$$

The decoupled $H(s)$ is as follows:

$$
H(s) = \begin{bmatrix} r+2Ls & 0 & 0 & 0 \\ 0 & r+2Ls & 0 & 0 \\ 0 & 0 & r+2(L-M)s & 0 \\ 0 & 0 & 0 & r+2(L+M)s \end{bmatrix} \tag{A7}
$$

At the same time, the matrix $I_L$ and $d$ are transformed.

$$
I_L(s) = PI_n(s) \tag{A8}
$$

$$
d(s) = Pu(s) \tag{A9}
$$

Substituting Formulas (A4), (A8), (A9) into Formula (A1), the result is as Formula (A10):

$$(v_1 - v_o)P^{-1} \begin{bmatrix} 1 \\ 1 \\ 1 \\ 1 \end{bmatrix} + v_o \begin{bmatrix} u_1 \\ u_2 \\ u_3 \\ u_4 \end{bmatrix} = HI_n \tag{A10}$$

Resolve the matrix form, as shown in the Formula group (A11).

$$\begin{cases} u_1 v_o = (r + 2Ls)i_{n1} \\ u_2 v_o = (r + 2Ls)i_{n2} \\ u_3 v_o = [r + 2(L - M)s]i_{n3} \\ v_1 - (1 - u_4)v_o = [r + 2(L + M)s]i_{n4} \\ C\frac{dv_o}{dt} = -2u_1 i_{n1} - 2u_2 i_{n2} - 4u_3 i_{n3} + 4(1 - u_4)i_{n4} - \frac{v_o}{R_L} \end{cases} \tag{A11}$$

When $i_{L1}^* = i_{L2}^* = i_{L3}^* = i_{L4}^*$, $i_{n1}^* = i_{n2}^* = i_{n3}^* = 0$ and $i_{n4}^* = i_{L4}^*$, Formula (A12) can be transformed into Formula (A13).

$$C\frac{dv_2}{dt} = -2u_1 i_{n1} - 2u_2 i_{n2} - 4u_3 i_{n3} + 4(1 - u_4)i_{n4} - \frac{v_2}{R_L} \tag{A12}$$

$$\frac{C}{4}\frac{dv}{dt} = (1 - u_4)i_{n4} - \frac{v_2}{4R_L} \tag{A13}$$

(2)   The small signal model near the steady-state is as follows:

$$G_{iu1} = G_{iu2} = \frac{V_o}{r + 2Ls} \tag{A14}$$

$$G_{iu3} = \frac{V_o}{r + 2(L - M)s} \tag{A15}$$

$$G_{iu4} = \frac{V_o\left(Cs + \frac{2}{R_L}\right)}{2C(L + M)s^2 + \left(Cr + \frac{2(L+M)}{R_L}\right)s + 4U_4 2} \tag{A16}$$

$$G_{v2in4} = \frac{4u_4 2R_L - 2(L + M)s - r}{(Cs + \frac{2}{R_L})R_L u_4} \tag{A17}$$

The current transfer function after decoupling is shown in Figure A1.

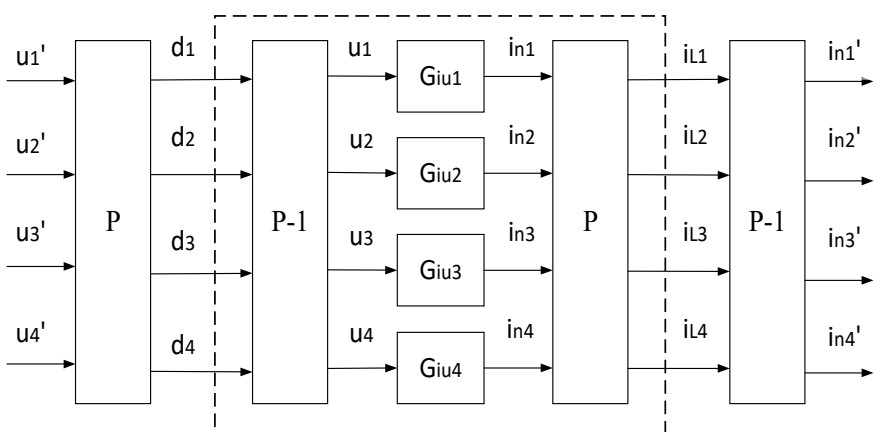

**Figure A1.** Block diagram of decoupling current loop control.

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
