# Peer review of "Modeling of Ship DC Power Grid and Research on Secondary Control Strategy"

_jmse, doi:10.3390/jmse10122037_

Round 1
Reviewer 1 Report
This paper deals with the modeling of ship DC power grid and research on secondary control strategy.
The topic is interesting, and this paper is ambitious, with a strong desire to consider important aspects in the field of ship DC power grid planning. The content of the manuscript is good, and both organization and presentation are satisfactory.
However, the paper requires revisions before it can be accepted.
Ship DC power grid has attracted the interest of many researchers, and many approaches have been proposed so far for the same topic. So, authors must show a clear contribution. Authors should cover the latest findings in this area where the contribution was done so that we can see what the improvement is compared to existing works.
Author Response
Point 1: The topic is interesting, and this paper is ambitious, with a strong desire to consider important aspects in the field of ship DC power grid planning. The content of the manuscript is good, and both organization and presentation are satisfactory.
However, the paper requires revisions before it can be accepted.
Ship DC power grid has attracted the interest of many researchers, and many approaches have been proposed so far for the same topic. So, authors must show a clear contribution. Authors should cover the latest findings in this area where the contribution was done so that we can see what the improvement is compared to existing works.
Response: Thank you for your advice. It has been added. “In previous distributed control strategies, in order to compensate voltage drop, devices need to exchange their own voltage information with adjacent devices. To achieve accurate current sharing, the equipment needs to exchange its own current information with adjacent equipment. In order to balance the state of charge of the battery, the device needs to exchange one or two messages with adjacent devices. In the control strategy proposed in this paper, the device only needs to exchange one or two messages with adjacent devices.”
Reviewer 2 Report
The research theme and method are very interesting. However the paper structure and the description are not clear. This led to the reader to read many times the text to understand the authors contribution. This penalty the paper in terms of impact.
My suggestion is to carefully rewrite the paper (mainly the first part, including the introduction) by using more clear sentences. Please clearly indicate what you want to demonstrate, the method used to demonstrate it and finally how you verify that results agree with your expectation.
Starting from the abstract, the authors have not clearly defined the problem. As example, why the distributed control is preferred over the centralized one in a shipboard (line 10)?
How the author understand that the simulation results agree with the reality and therefore the models are accurate (line 16)?
Moving to the introduction, the references are good and clearly define the problem, but it is very difficult to understand from the paper introduction the state of the art. In this way a reader must read the references to have his personal idea of the proposed problem. Please add some sentences to clearly indicate what is the problem that you are defining, and what is the contribution of each reference paper. Please reformulate also some sentences to improve the readability.
Section 2: Is the interleaved mathematical model (1) new and written by authors? If not please report the reference papers.
The same for the subsection 2.2. Is the control scheme common or defined by authors? If not new, pleas report references. From Figure 2, it is not clear why only the channel 4 is used for the voltage control, while remaining current control have zero as reference. Why the authors have not equally divided the reference current between phases?
Line 128. The authors starting from the introduction cite the droop curve. In the Section 3 use this parameter but in the paper the parameter is not clearly described. Could you improve this part?
Line 175. The eq. (20) depends by the line and loads parameters. What happen in case of parameter changes? What is the stability limit? Is the linearization method good for this grid? Why the Lyapunov method is not used instead of the linearized pole placement? The authors should clearly explain why the proposed method is enough for the application.
Author Response
Point 1. The research theme and method are very interesting. However the paper structure and the description are not clear. This led to the reader to read many times the text to understand the authors contribution. This penalty the paper in terms of impact.
My suggestion is to carefully rewrite the paper (mainly the first part, including the introduction) by using more clear sentences. Please clearly indicate what you want to demonstrate, the method used to demonstrate it and finally how you verify that results agree with your expectation.
Starting from the abstract, the authors have not clearly defined the problem. As example, why the distributed control is preferred over the centralized one in a shipboard (line 10)?
Response: Centralized control and distributed control have their own advantages. We just want to express that the distributed structure is more stable.
How the author understand that the simulation results agree with the reality and therefore the models are accurate (line 16)?
Response: The mathematical model adopts the model in this paper.” [10] Bing Su. Research on topology and control of multi-phase interleaved bidirectional DC/DC converter with coupled inductors. Shandong University, 2020.”
We mainly refer to this paper, in which the theory is in good agreement with the experiment,
Moving to the introduction, the references are good and clearly define the problem, but it is very difficult to understand from the paper introduction the state of the art. In this way a reader must read the references to have his personal idea of the proposed problem. Please add some sentences to clearly indicate what is the problem that you are defining, and what is the contribution of each reference paper. Please reformulate also some sentences to improve the readability.
Response: It has been added. “In previous distributed control strategies, in order to compensate voltage drop, devices need to exchange their own voltage information with adjacent devices. To achieve accurate current sharing, the equipment needs to exchange its own current information with adjacent equipment. In order to balance the state of charge of the battery, the device needs to exchange one or two messages with adjacent devices. In the control strategy proposed in this paper, the device only needs to exchange one or two messages with adjacent devices.”
Point 2: Is the interleaved mathematical model (1) new and written by authors? If not please report the reference papers.
The same for the subsection 2.2. Is the control scheme common or defined by authors? If not new, pleas report references. From Figure 2, it is not clear why only the channel 4 is used for the voltage control, while remaining current control have zero as reference. Why the authors have not equally divided the reference current between phases?
Response: It has been added.” Bing Su. Research on topology and control of multi-phase interleaved bidirectional DC/DC converter with coupled inductors. Shandong University, 2020.”
Because There is coupling between inductors. When,.
Point 3: Line 128. The authors starting from the introduction cite the droop curve. In the Section 3 use this parameter but in the paper the parameter is not clearly described. Could you improve this part?
Response: It has been added. Droop coefficient Rv is set to 0.5.
Point 4: Line 175. The eq. (20) depends by the line and loads parameters. What happen in case of parameter changes? What is the stability limit? Is the linearization method good for this grid? Why the Lyapunov method is not used instead of the linearized pole placement? The authors should clearly explain why the proposed method is enough for the application. Please find the comments to your manuscript:
Response:
Because 0<k<1, so .G(s)»1. So, the coefficients of the zeroth term and the first term of s are positive numbers

Reviewer 3 Report
Please find the comments to your manuscript:
1. Please match the symbols in Equation (1) and Figure 1. and add explanation of v1, M, r, vo, c etc..
2. What are Rdg and R1,2,3 in Figure 3? Please add an explanation so the reader can understand it.
3. Please add the name of the y-axis in Figure 6.
4. I suggest changing the order of Figure 8.
(a) Output voltage w/o SC (b) Output current w/o SC (c) DC bus voltage.
5. I recommend enlarging the picture of the transient state in Figure 8.
6. What is the result in the simulated condition of Figure 8 in the proposed control? In order to confirm the difference between the conventional control and the proposed control, comparison of simulation results under the same conditions is required.
7. In Figure 9, why is the output of the generator slowly increased over 20 seconds? What is the result of simulation for sudden power fluctuation?
8. In Figure 9 (d), the DC link voltage is not constantly controlled. If the DC link is not constant, it may affect the stable operation of the load system. Explain why.
Author Response
Point 1. Please match the symbols in Equation (1) and Figure 1. and add explanation of v1, M, r, vo, c etc.
Response: Thank you for your advice. It has been added.” is the voltage at the low voltage side of the converter, is the voltage at the high voltage side of the converter, L is the inductance, M is the mutual inductance, r is the resistance, and C is the output capacitance of the converter.”
Point 2. What are Rdg and R1,2,3 in Figure 3? Please add an explanation so the reader can understand it.
Response: Thank you for your advice. It has been added.” 、R1、R2…Rn are the line impedance of generator and batteries respectively.”
Point 3. Please add the name of the y-axis in Figure 6.
Response: Figure 6 shows the convergence of the algorithm. The numerical value has no physical significance.
Point 4. I suggest changing the order of Figure 8.
- Output voltage w/o SC (b) Output current w/o SC (c) DC bus voltage.
Response: Figure 8. Simulation without secondary control: (a); Output voltages of converters (b) Output currents of converters; (c) DC Bus Voltage.
Point 5. I recommend enlarging the picture of the transient state in Figure 8.
Response:
Point 6. What is the result in the simulated condition of Figure 8 in the proposed control? In order to confirm the difference between the conventional control and the proposed control, comparison of simulation results under the same conditions is required.
Response: Figure 8 shows an inherent contradiction of droop control. Selecting smaller droop curve coefficient can reduce the voltage deviation, but will reduce the accuracy of shared current; Selecting a larger droop curve coefficient can improve the accuracy of the shared current, but will increase the voltage deviation. To avoid voltage deviation, load current can not be accurately shared and other problems. Secondary control is adopted.
Point 7. In Figure 9, why is the output of the generator slowly increased over 20 seconds? What is the result of simulation for sudden power fluctuation?
Response: The output power of the generator is increased to show the effect of the control strategy when the battery is charged
Point 8. In Figure 9 (d), the DC link voltage is not constantly controlled. If the DC link is not constant, it may affect the stable operation of the load system. Explain why.
Response: The proposed strategy is a distributed control strategy. One of the objectives of the control is that the average value of the output voltages of converters is equal to its rated value. Due to the line impedance, it is normal for the voltage at both ends of the load to slightly deviate from the rated voltage, which will not affect the bus voltage stability.

Reviewer 4 Report
The DC network on large ships is not used. The article is theoretical. The article is well written. The results will not find practical application. It can be published as a curiosity about the possibility of using a DC network.
Author Response
Point 1. The DC network on large ships is not used. The article is theoretical. The article is well written. The results will not find practical application. It can be published as a curiosity about the possibility of using a DC network.
Response: Thank you for your advice.
Response: Practical Cases of DC-SMG
|
Case(Year) |
Type |
Company |
Power |
Capacity |
Voltage |
Detail information |
|
Royal Princess(2012) [1] |
Cruise ship |
Fincantieri Shipyard Italy |
78MVA |
N/A |
11kV |
Generators:2´21MW+2´18MW |
|
Dina Star(2013)[141] |
PSV |
Kleven Maritime Norway |
10MW |
N/A |
1kv |
Generators:4´2.35MW+1´0.97MW |
|
Edda Ferd(2013) [142],[143] |
PSV |
Qstensjq Rederi AS Norway |
11MW |
338kwh |
700V |
Generators:2´2.2MW+2´3.3MW+158kW; Batteries:52´6.5kwh. |
|
Fannefjord(2014)[144],[145] |
Ferry |
Fjord1,Norway |
2.8MW |
410kWh |
1050V |
Two LNG engines (2´900kW),one diesel engine(1000kW) and battery hybrid system. |
|
ReVolt(2014)[146] |
Short-sea vessel |
DNV GL |
N/A |
3MWh |
N/A |
Autonomous and unmannned,fully battery powered |
|
Ampere(2015)[147] |
Car ferry |
Norwegian Shipyard Fjellstrand |
900kW |
1MWh |
810-1050V |
All-electric-ship. Batteries;2´520kWh. |
|
M/F Tycho Brahe,M/F Aurora(2017)[148],[149] |
Ferry |
HH Ferries Group Denmark |
11MW |
4160kWh |
1kV |
Gensets:4´2.6MW; Lithium batteries:640´6.5kWh. |
|
Future of The Fjords(2018)[150] |
Catamaran |
The Fjords DA |
1.8MW |
1.8MWh |
1kV |
All-electric-ship.Batteries:2´900kWh. |
|
USS Zumwalt DDG-1000 destroyer(2008)[151] |
Destroyer |
US Navy |
78MW |
N/A |
4160V |
Prime movers:2´35.4MW+2´3.8MW |
Reviewer 5 Report
Compare your result with any hardware simulation software and discuss the robustness of the proposed method
Author Response
Point 1. Compare your result with any hardware simulation software and discuss the robustness of the proposed method.
Response: In consideration of time, this technology has not been implemented on the board or hardware simulation software, but this work will be carried out in the follow-up study.
Round 2
Reviewer 1 Report
The paper is improved, and issues are addressed.
Reviewer 3 Report
Reflection of the reviewer's opinion is appropriate.